# Transcription-Driven Repurposing of Cardiotonic Steroids for Lithium Treatment of Severe Depression

**DOI:** 10.3390/cells14080575

**Published:** 2025-04-11

**Authors:** Richard Killick, Claudie Hooper, Cathy Fernandes, Christina Elliott, Dag Aarsland, Svein R. Kjosavik, Ragnhild Østerhus, Gareth Williams

**Affiliations:** 1Centre for Healthy Brain Aging, IoPPN, King’s College London, London SE5 9RT, UK; 2IHU HealthAge, Gérontopôle, Department of Geriatrics, CHU Toulouse, 31059 Toulouse, France; 3Social, Genetic & Developmental Psychiatry Centre, IoPPN, King’s College London, London SE5 8AF, UK; 4MRC Centre for Neurodevelopmental Disorders, IoPPN, King’s College London, London SE1 1UL, UK; 5Faculty of Medical Sciences, School of Biomedical, Nutritional and Sport Sciences, Newcastle University, Newcastle NE4 5TG, UK; 6Centre for Age-Related Medicine (SESAM), Stavanger University Hospital, 4011 Stavanger, Norway; 7General Practice and Care Coordination Research Group, Stavanger University Hospital, 4011 Stavanger, Norway; 8Wolfson SPaRC, IoPPN, King’s College London, London SE1 1UL, UK

**Keywords:** drug repurposing, differential gene expression, epidemiological analysis, major depressive disorder, bipolar disorder, lithium, cardiotonic steroids

## Abstract

Lithium is prescribed as a mood stabilizer for bipolar disorder and severe depression. However, the mechanism of action of lithium is unknown and there are major side effects associated with prolonged medication. This motivates a search for safer alternative drug repurposing candidates. Given that the drug mechanism may be encoded in transcriptional changes, we generated the gene expression profile for acute lithium treatment of cortical neuronal cultures. We found that the lithium-associated transcription response harbors a significant component that is the reverse of that seen in human brain samples from patients with major depression, bipolar disorder, and a mouse model of depression. Interrogating publicly available drug-driven expression data, we found that cardiotonic steroids drive gene expression in a correlated manner to our acute lithium profile. An analysis of the psychiatric medication cohort of the Norwegian Prescription Database showed that cardiotonic prescription is associated with a lower incidence of lithium prescription. Our transcriptional and epidemiological observations point towards cardiotonic steroids as possible repurposing candidates for lithium. These observations motivate a controlled trial to establish a causal connection and genuine therapeutic benefit in the context of depression.

## 1. Background

Despite the long therapeutic pedigree of lithium (Li) as a mood stabilizer [1], its mechanism of action is still unclear and use is associated with major side-effects [2]. The side effects are likely due to the non-specific effects of Li covering the modulation of serotonin [3,4], dopamine [5], and glutamate [6] neurotransmitters, and the inhibition of glycogen synthase kinase-3 (GSK-3) [7] and inositol monophosphatase [8,9]. Li can also enter through sodium channels [10], block potassium channels [11], and impact calcium mobilization [12], subsequently modulating neuronal ion channel activity and membrane potential [13]. The side-effect burden [2] motivates a search for an alternative medication, and the wide spectrum of activity points towards a candidate emerging based on a global phenotype comparison. In particular, a drug repurposing strategy, where existing therapeutics emerge as candidates for conditions for which they were not originally developed, is a potential avenue based on the availability of extensive activity data from laboratory experiments and prescription histories [14]. One insight into the mechanism can be obtained via global transcription profiling [15,16,17]. A direct validation of drug-associated transcription encoding therapeutic activity would be the observation that the gene expression changes driven by the drug are reverse of those seen in disease states [18,19]. This transcription-based drug repurposing approach has been used in the search for alternative candidates in cancer therapy [18,20] and neurodegenerative disease [14,21,22]. Mapping biological activity to transcriptional perturbation is a particularly fertile option as the data are both quantitative and multidimensional with an abundance of publicly available data on disease state profiles, through online depositories such as the National Center for Biotechnology Information (NCBI) Gene Expression Omnibus (GEO) [23], and drug-associated profiles via the Connectivity Map (CMAP) [15] and Library of Integrated Network-Based Cellular Signatures (LINCS) [24] datasets. One can also base repurposing on actual epidemiological data where disease incidence is tracked in relation to drug candidate use. A recent notable example is the repurposing of salbutamol for Parkinson’s disease (PD) [25], where the incidence of PD over a five-year period was shown to decrease in proportion to salbutamol, a beta-adrenergic agonist, prescription and increase with the prescription frequency of a beta-adrenergic antagonist, propranolol. The obvious drawback of this methodology is that retrospective study analysis can at best point to causality and serve to either confirm hypotheses or motivate further investigation or controlled trials. In the present work, we sought to delimit repurposing candidates for Li on the basis of an expression profile correlation and go some way to validate the candidates based on a correlation analysis of prescription data, with the hypothesis that those taking the candidate drug are less likely to be prescribed Li.

We first sought to define an activity profile for Li. We reasoned that the most likely activity representation would be furnished by gene expression profiling, as there is a wealth of data on compound-driven gene expression changes [15,24]. We also wanted to focus on the immediate downstream effects of Li treatment as these will more likely reflect the drug’s direct engagement with primary targets and avoid confounding effects due to compensatory changes. Thus, our investigation will furnish a valuable addition to the existing studies of the Li-driven transcriptional landscape [26,27,28,29,30,31]. Assuming the therapeutic effects are primarily driven by neuronal interactions, we profiled Li in primary neuronal cultures. We reasoned that these primary neuronal cultures provide a better approximation to the disease context drug activity than cell lines and, in contrast to in vivo assays, allow for a precise control of dosing and treatment time. To investigate the therapeutic aspects of the gene expression changes driven by Li, we analyzed transcriptional data on two conditions for which Li is commonly prescribed: major depressive disorder (MDD) and bipolar disorder (BP). There is a wealth of gene expression data on both these conditions: MDD [32,33,34,35,36] and BP [35,37,38,39,40,41,42,43,44]. We report here that our acute Li profile (ALP) shows a significant negative correlation with transcriptional profiles corresponding to both MDD and BP derived from post-mortem brain samples. Interestingly, we also found a negative correlation with a chronic variable stress (CVS) [45] mouse model brain profile [33]. This observation bolstered the hypothesis that ALP captures some of the therapeutic aspects of Li activity and motivated our search for compounds with similar transcriptional profiles as possible repurposing candidates. We found that multiple cardiotonic steroids (CTS) significantly correlate with the ALPS, MDD, and CVS profiles.

There are two CTS currently in the clinic: digoxin and digitoxin. We reasoned that if CTS use has a therapeutic component in common with Li, then there may be fewer prescriptions of Li in those on CTS medication. To investigate this possibility, we gathered data from the Norwegian Prescribed Drug Registry (NorPD, www.norpd.no, 22 July 2024) that consist of prescription data across the public health sector from 2004 onwards. We decided to focus our analysis exclusively on individuals who have been prescribed at least one form of psychiatric medication, thus aiming to better isolate the specific factors influencing the decision to prescribe lithium. This restriction ensures that our study population has already engaged with psychiatric care at a level that necessitates pharmacological intervention, hence providing a more relevant and homogeneous baseline for comparison. Including individuals without any psychiatric medication history might introduce confounding variables related to barriers to care, varying diagnostic thresholds, or treatment-seeking behaviors, which could dilute or obscure the factors specifically associated with lithium prescription. This cohort still comprises a significant fraction of the entire population (2 million of 5.5 million). To this end, we compiled prescription records for 94 psychiatric medications and two CTS (digoxin and digitoxin) for the entire Norwegian population covering the years 2010 to 2021. Our analysis points to a significant reduction in Li prescriptions for those on CTS medication.

## 2. Methods

### 2.1. Cell Culture Treatment

All materials were obtained from Sigma (St. Louis, MO, USA) unless otherwise specified. Time-mated Sprague Dawley rats were from Charles River (Wilmington, MA, USA). Neurobasal medium, B27 supplement, and Trizol were from Invitrogen (Waltham, MA, USA). Papain dissociation systems were from Worthington (Worthington, OH, USA). Streptavidin-phycoerythrin was from Molecular Probes (Eugene, OR, USA); Biotinylated anti-streptavidin antibody from Vector Laboratories (Newark, CA, USA); Taqman reverse transcription (RT) reagent kit from Applied Biosystems (Waltham, MA, USA); QuantiTect SYBR green kit from Qiagen (Venlo, The Netherlands); and Rat Genome 230 v2.0 whole genome microarray chips and the one cycle target labeling and control kit were from Affymetrix (Santa Clara, CA, USA).

Primary cortical neuronal cultures were generated from Sprague Dawley E18 rat embryos using papain dissociation according to manufacturers’ instructions. Neurons were cultured in a neurobasal medium supplemented with B27 for 7 days at 37 °C in a humidified atmosphere of 5% CO_2_ in the air as previously described [46]. After 7 days in culture (d.i.c.), neurons were washed twice in neurobasal medium (without any supplements) and then left in fresh neurobasal medium for 2 h, to remove insulin, a constituent of B27 supplement and an inhibitor of GSK3. Following this incubation period, LiCl was then added to the media at a final concentration of 10 mM, and cells were cultured for a further 2 h before collection. This dose is a widely used sub-toxic acute dose that has rapid effects on targets including GSK3 and which mimics Wnt signaling activity [7] and impacts the NR4A family [47].

### 2.2. Microarray Data

Total RNA was extracted from primary rat cortical neurons 2 h post-treatment using Trizol followed by isopropanol precipitation. The RNA integrity number (RIN), a measure of RNA quality (10 being maximal), ranged from 9.6–10 for all samples, as determined using an Agilent Bioanalyser (Santa Clara, CA, USA) Total RNA (7.5 mg) was reversed transcribed using oligo (dT), biotinylated, and hybridized to Affymetrix Rat Genome 230 v2.0 whole genome microarrays using the one cycle target labeling and control kit according to protocols in the Affymetrix GeneChip Expression Analysis Technical Manual. The arrays were washed and stained with streptavidin-phycoerythrin and the fluorescent signal was amplified with a biotinylated anti-streptavidin antibody. Staining and washing were performed using an Affymetrix fluidics system and microarrays were scanned using an Affymetrix gene chip 3000 scanner (Santa Clara, CA, USA). The full expression profile for Li is given in Appendix A.

### 2.3. Expression Profile Analysis

The CEL file data were RMA normalized using the affy package in the Bioconductor R environment (http://www.bioconductor.org/, 6 April 2025). Differential gene expression profiles for the lithium treatment performed in our laboratory and for the data sourced from NCBI GEO were generated using the R limma package (version 3.20). To facilitate expression profile comparison and generate composite profiles the gene expression changes were converted to Z score values and composite profiles were defined as the summed Z scores divided by the square root of the number of experiments for each gene. In the meta-analysis of the disease profiles, we compared with brain region, sex, and age where these features were provided.

Repurposing candidates were defined based on a global transcription correlation analysis where the query profile was scored against the CMAP [15] repository of drug-driven differential expression profiles, consisting of 1309 drug-like compounds.

### 2.4. Publicly Available MDD and BPD Gene Expression Data

Interrogating the NCBI GEO gene expression repository, we found five expression series corresponding to brain samples from sufferers of major depressive disorder (MDD) with matched controls with GEO identifiers GSE101521 [34], GSE102556 [33], GSE53987 [35], GSE44593, and a study [36] split over the following series: GSE54562, GSE54563, GSE54564, GSE54565, GSE54567, GSE54562, GSE54568, GSE54570, GSE54561, GSE54572, GSE54575. This enabled us to delimit an MDD meta-profile based on combining the Z scores for gene expression changes relative to control groups from the various independent datasets. In cases where donor measures were included in the data, such as age, sex, brain region, and suicide status, these were factored into the differential expression analysis by being included as covariates in the linear model. Similarly, we generated a bipolar disorder (BPD) meta-profile from six independent series (GSE5392 [37], GSE53239 [38], GSE53987 [35], GSE62191 [39], GSE80336 [40], and GSE81396 [41]). Our meta-analysis also included a differential expression profile for a mouse chronic unpredictable stress (CUS) model of depression [45] GSE102556 [33]. See Appendix A for the summary profiles.

### 2.5. NorPD Prescription Data

Prescription data consisting of prescription date, date of birth, date of death in case of death were obtained from the NorPD prescription database for 95 psychiatric medications (802,109 men and 1,123,759 women) and two commonly prescribed cardiotonics, digoxin and digitoxin (24,099 men and 23,996 women and 14,994 men and 17,890 women of these in the psychiatric medication cohort), see Appendix A for the Anatomical Therapeutic Chemical (ATC) codes. Prescription associations were assessed through logistic regression with appropriate covariates, using the R glm() function.

## 3. Results

### 3.1. Transcriptional Changes Associated with Acute Li Treatment

On the premise that knowing which genes are responsive to Li could shed light on the mechanism of action, we examined the transcriptional effects of Li on rat primary cortical neurons using whole genome expression microarrays. A number of microarray-based expression studies, performed in model systems ranging from yeast through neuronal cell lines to brains of treated rodents, have been reported. A considerable number of gene expression studies have been undertaken in varied model systems with the aim of shedding light on the mechanism of action of Li on mood [48,49,50,51,52,53]. All employed chronic treatment regimens given by Li require from several weeks to up to six months to show maximal clinical benefit in men. We reasoned that such chronic treatments could miss the acute, “first wave”, effects of Li on gene expression due to masking by subsequent waves of transcription. In light of this, we measured the transcription changes two hours post-treatment of rat cortical neuronal cultures.

We found that a disproportionate number of the genes regulated by the acute treatment encode for transcription factors; thus, we believe we have uncovered, for the first time, the direct effects of Li on neuronal gene transcription. In total, 2157 genes (643 UP 1514 DOWN) showed expression changes at the *p* < 0.05 significance level, see Figure 1 for a list of the most highly regulated genes. In addition to the regulation of transcription factors notable amongst the upregulated genes implicated in neuronal function are the brain-derived neuronal growth factor (BDNF), all three members of the nuclear receptor subfamily 4A (NR4A) of genes involved in synaptic mitochondrial function, and calcium-dependent kinase kinase 2 (CAMKK2) involved in learning and memory.

### 3.2. The Li Expression Profile in the Context of Depression

It is known that depression is associated with consistent expression changes in the brains of sufferers, see for example Pantazatos SP et al. [34], and this motivated us to investigate the relationship of depression-associated expression changes with our acute Li expression profile. Our hypothesis is that if a component of biological activity is encoded in the perturbation of gene expression, we would expect the signature of a therapeutic to have a component that is reverse of that seen in the corresponding disease state. Interrogating the NCBI GEO repository, we found five patient cohorts corresponding to brain samples from sufferers of MDD, see Section 2 (Methods) for details. It is apparent that the genes most responsive to Li treatment tend to be perturbed in the reverse sense in MDD, see Table 1, and to a lesser extent in BPD, see Table 2. Of interest is that a CVR mouse model of depression shows a consistent expression profile with MDD. The mouse model data were sourced from GSE102556 [33]. Further, the expression changes seen in CVR are likewise reverse of those driven by Li, see Table 2. The combined BPD profile shows a smaller but still significant anti-correlation with ALP, see Table 2. This opens up the interesting possibility of validating intervention candidates in the mouse by tracing the expression changes driven by therapeutic candidates.

### 3.3. Repurposing Candidates

Transcription provides a rational route to repurposing with the hypothesis that drugs reversing the gene expression changes associated with disease states may ameliorate the condition [14,15,19]. Further, provided that an existing drug’s activity is encoded in transcription [15,24], this can facilitate a target for sourcing alternative therapeutics. As shown above, both Li treatment of cortical cultures and the MDD depression state in the brain are associated with substantial transcriptional changes. Further, bolstering the anti-depressive effects of Li, the expression changes driven by Li are significantly anti-correlated with those seen in MDD brain-derived samples. This led us to adopt a transcription-based repurposing strategy for Li and look for alternative drugs with transcription profiles correlating with the Li profile and also showing an anti-correlation with the MDD expression changes. To this end we performed queried the CMAP set of 1309 drug-like compound-driven expression profiles. As can be seen in Table 3, we found eight cardiotonic steroids both significantly correlate with the Li profile and anti-correlate with the MDD summary profile.

### 3.4. Prescription Frequencies of CTS and Li in the NorPD Psychiatric Medication Cohort

We have motivated the repurposing of CTS as alternative to Li treatment in severe depression based on shared features in gene expression perturbation. It is then of interest to see to what extent the prescription of CTS is associated with a reduced likelihood of being prescribed Li. To focus on the specific factors leading to Li prescription, we decided to restrict our analysis to those being prescribed at least one form of psychiatric medication, see Methods for medications. This cohort comprises a significant fraction of the entire population (2 million of 5.5 million). To this end, we compiled prescription records for 94 psychiatric medications and 2 CTS (digoxin and digitoxin) for the entire Norwegian population covering the years 2010 to 2021, see Section 2 (Methods). As can be seen from Table 4, there is a reduced Li prescription frequency in those prescribed cardiotonic steroids. Performing a logistic regression with age and sex as covariates, we obtain a log odds association of Li with cardiotonic steroid use of −1.15 (95% CO [−0.94 −1.37] *p* = 6.24 × 10^−27^). Here, medication status is binary with prescription status assigned to those with at least one prescription of the given medication. To test the robustness of the negative association we varied the minimal prescription count required for positive prescription status, see Table 4. We also varied the cohort’s minimal age and restricted analysis based on sex. We observed a consistent negative association of cardiotonic use with Li prescription, see Table 4. This observation bolsters our transcriptional analysis.

## 4. Conclusions

The compromising side-effects and poorly understood mechanism of action of the widely prescribed Li in the context of major depression motivated our investigation into the possibility of repurposing drugs with established safety profiles that might be alternatives to Li treatment. We sought to delimit the Li activity mechanism through an analysis of acutely driven gene expression changes in the context of Li treatment of neuronal cells. Our lithium profile, ALP, showed an upregulation of key genes involved in neuronal function. Notably, brain-derived neurotrophin, BDNF, genes involved in synaptic mitochondrial function, NR4A, and learning and memory, CAMKK2. We also found that ALP shows significant anti-correlation with human brain sample-derived MDD and BP profiles. These observations led us to investigate the potential of ALP to capture repurposing candidates from databases of drug-driven expression profiles. Querying CMAP with ALP we found that cardiotonic compounds constituted high-rank hits. Interestingly, cardiotonics also showed a significant anti-correlation with MDD and the mouse model CVS profiles. Bolstered by these observations we investigated whether prescribed CTS are associated with a lower Li prescription rate. Our analysis of the NorPD prescription data for those on psychiatric medication showed that CTS use is associated with a lower likelihood of Li prescription.

These observations motivate further investigation of the potential of CTS as an alternative to Li in the context of severe depression. However, the results presented here are only in the form of correlation analyses and do not establish a causal basis for the novel therapeutic potential of CTS. It would be of interest, for example, to see to what extent CTS reverses behavioral changes in the CVS mouse model of depression. The epidemiological analysis showing a decreased prescription of Li in those on CTS medication goes some way in validating CTS as a candidate. Such retrospective observations though have sometimes led to disputed claims, such as the proposed protective effects of statins in cancer [54]. A retrospective analysis of NorPD prescription data provided strong evidence for the protective effects of salbutamol against PD [25]. Subsequent analysis of United States Medicare data controlling for predicted smoking incidence found that the negative salbutamol association with PD incidence was no longer observedd [55]. They argue that salbutamol use is higher in smokers and smoking reportedly lowers PD risk [56]. Interestingly, a transcriptional approach to repurpose drugs on the basis of their driving growth factor expression in the brain and subsequently providing trophic support for neurons in the context of PD led to the identification of salbutamol as a potential candidate, which was shown to reverse dopaminergic loss in a mouse model of PD [22]. As with any proposed repurposing, the relative safety profiles of the existing and candidate drugs have to be carefully considered before deployment [57].

It has been reported that heart disease can be a co-morbidity of clinical depression, with cardiovascular mortality being higher in those with depression [58]. This can be explained on the basis of lifestyle changes associated with depression or could have a more direct physiological connection as both conditions can be characterized by increased inflammation [59,60] and autonomic nervous system dysregulation may lead to cardiac problems [61]. These observations would point towards a higher prescription rate of heart medications in the major depression cohort, and it would be of interest to contrast the prescription rates of CTS with other heart medications. In conclusion, our analysis suggests a possible role for CTS as a Li replacement in MDD and motivates further investigation in the form of a controlled trial.

## Figures and Tables

**Figure 1 cells-14-00575-f001:**
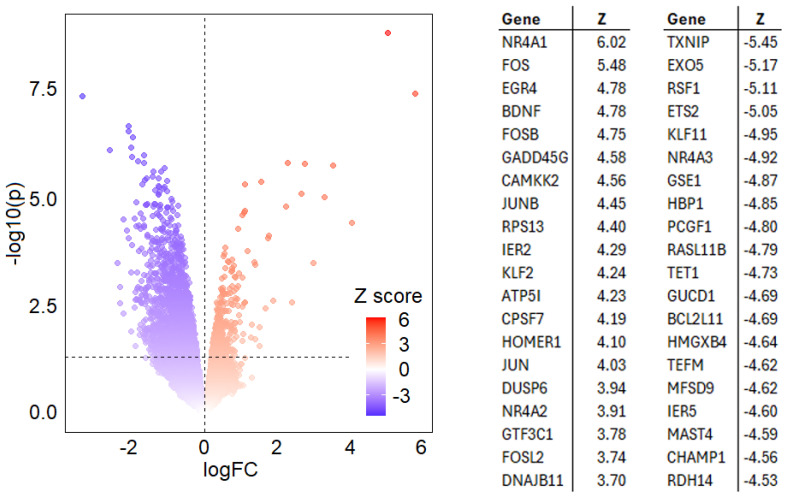
Acute lithium expression response. Lithium elicits a substantial transcriptional response two hours post-treatment of cortical neurons differentiated from embryonic rat progenitor cells. In total, 2157 genes (643 UP 1514 DOWN) showed expression changes at the *p* < 0.05 significance level. The volcano plot is shown left, colored according to Z score, and the top-up and downregulated genes are shown on the right.

**Table 1 cells-14-00575-t001:** Li shows substantial reversal of expression changes in MDD and BPD. The top-most regulated genes are shown together with the expression changes in the combined meta-analysis profiles corresponding to MDD (left) and BPD (right) showing a clear reversal. Of the 40 most regulated genes, due to Li treatment, 32 in 40 and 22 in 32 are regulated in an opposite sense in MDD and BPD. The entries are coloured according to Z score value.

GENE	ALP	MDD	GENE	ALP	MDD
NR4A1	6.02	−2.72	TXNIP	−5.45	2.24
FOS	5.48	−2.95	ETS2	−5.05	−4.04
EGR4	4.78	−4.53	KLF11	−4.95	2.51
BDNF	4.78	−3.82	NR4A3	−4.92	−2.46
JUNB	4.45	−2.27	GSE1	−4.87	−2.82
RPS13	4.40	2.06	BCL2L11	−4.69	2.93
IER2	4.29	−2.16	IER5	−4.60	−2.15
HOMER1	4.10	−2.71	FBXL3	−4.44	2.01
DUSP6	3.94	−4.97	KLF15	−4.29	2.70
NR4A2	3.91	−3.93	CREBRF	−4.24	2.44
FOSL2	3.74	−4.11	PLXNA2	−4.20	−4.06
CREM	3.68	−3.21	PIK3IP1	−4.16	3.38
SIK1	3.59	−2.96	ZKSCAN1	−4.14	3.95
SRF	3.54	−2.19	DSTYK	−4.06	3.90
RGS2	3.37	−2.48	ZEB2	−4.01	2.72

**Table 2 cells-14-00575-t002:** There is a consistent anti-correlation between the acute lithium response profile and gene expression changes seen in MDD BPD and a mouse model of depression. There is a significant anti-correlation of the acute lithium profile with the combined MDD and BPD profiles, the correlations are given as Pearson correlation coefficients with 95% confidence intervals and associated null probabilities. Interestingly, a mouse model of depression shows similarities with both the MDD and BPD profiles as well as a significant anti-correlation with the lithium profile.

PROFILE	Lithium	MDD COMBO	BPD COMBO
MDD COMBO	−0.24 [−0.34 −0.13] 2.65 × 10^−5^		
BPD COMBO	−0.13 [−0.23 −0.03] 0.01133	0.21 [0.12 0.29] 8.18 × 10^−6^	
CUS Mouse	−0.25 [−0.32 −0.17] 2.46 × 10^−9^	0.38 [0.30 0.46] < 2.2 × 10^−16^	0.13 [0.05 0.21] 0.001

**Table 3 cells-14-00575-t003:** Cardiotonic steroid expression signatures correlate with the Li profile and anti-correlate with a summary MDD profile. The correlation Z scores for seven cardiotonic steroids from CMAP against the Li and MDD summary profiles are shown together with the relevant rank (positive and negative, respectively).

	Lithium		MDD	
Compound	Z	Rank	Z	Rank
digitoxigenin	5.22	12	−5.73	14
digoxin	5.2	13	−6	9
ouabain	4.27	21	−7.19	2
proscillaridin	4.26	22	−5.27	24
digoxigenin	3.82	35	−2.99	168
lanatoside C	3.43	46	−6.19	6
helveticoside	2.74	94	−4.95	31

**Table 4 cells-14-00575-t004:** CTS prescription is consistently associated with a lower Li prescription likelihood. The log odds and statistics are shown for the full NorPD data for those on psychiatric medication over the period 2010 to 2021. CTS use negatively correlates with Li prescription in the full cohort (ALL), in the cohort with prescription status assigned to those with 10 or more prescriptions for the drugs, in age-restricted cohorts, and separately for men and women.

Cohort	LogODDS [95% CO]	Prob	Li and CTS	Li	CTS	TOT
ALL	−1.15 [−0.94 −1.37]	6.24 × 10^−27^	88	17,488	33,234	1,926,252
Pr > 10	−0.96 [−0.63 −1.28]	6.82 × 10^−9^	37	10,607	15,784	1,901,972
>40 years	−0.96 [−0.75 −1.17]	4.55 × 10^−19^	88	14,231	33,177	1,525,892
>70 years	−1.03 [−0.78 −1.29]	2.61 × 10^−15^	60	4355	30,290	652,988
men	−1.18 [−0.88 −1.47]	1.07 × 10^−14^	44	7663	15,155	802,285
women	−1.14 [−0.84 −1.44]	6.06 × 10^−14^	44	9825	18,079	1,123,967

## Data Availability

Access to NorPD can be found at http://norpd.no, 22 July 2024. Access to data and linkage of registries require ethics approval and permission from the individual registries.

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
