# Peer review of "Transcription-Driven Repurposing of Cardiotonic Steroids for Lithium Treatment of Severe Depression"

_cells, 2025, doi:10.3390/cells14080575_

Round 1
Reviewer 1 Report
Comments and Suggestions for Authors
Accept after Minor Revisions
This manuscript is ready for publication in the Journal, Cells, after a minor but consequential revision in the Reference Section. The use of the medication, Lithium in manic-depressive disorders, presently known as bipolar mental disorders, is well-known.
As an expert in the use of Lithium in the diagnosis and treatment of mental disorders, the Reviewer commends this multinational collaboration for providing excellent evidence to repurpose the Use of Lithium with Cardiotonic Steroids for Major Depressive Disorder and Bipolar Disorder.
The crux of the matter in the use of Lithium for long term use actually resides in the gene transcription changes that are caused by Lithium. The authors have successfully provided the data to revise the textbook in the matter of Lithium. Thus, the Reviewer requests a minor revision in the references. The rationale is as follows,
From early on, we, as scientists, have shown that one of the mechanisms of the action of Lithium depends on the function of the neurotransmitter, serotonin. The authors cite a relevant reference for serotonin in the action of Lithium. However, another citation is needed because there is direct biochemical evidence for the action of Lithium on serotonin. The authors are required to add the relevant reference, Neuropharmacology. 1982 Jul;21(7):671-9. doi: 10.1016/0028-3908(82)90010-7. PMID: 6181426.
Author Response
Please see our comments in italics below
Accept after Minor Revisions
This manuscript is ready for publication in the Journal, Cells, after a minor but consequential revision in the Reference Section. The use of the medication, Lithium in manic-depressive disorders, presently known as bipolar mental disorders, is well-known.
As an expert in the use of Lithium in the diagnosis and treatment of mental disorders, the Reviewer commends this multinational collaboration for providing excellent evidence to repurpose the Use of Lithium with Cardiotonic Steroids for Major Depressive Disorder and Bipolar Disorder.
The crux of the matter in the use of Lithium for long term use actually resides in the gene transcription changes that are caused by Lithium. The authors have successfully provided the data to revise the textbook in the matter of Lithium. Thus, the Reviewer requests a minor revision in the references. The rationale is as follows,
From early on, we, as scientists, have shown that one of the mechanisms of the action of Lithium depends on the function of the neurotransmitter, serotonin. The authors cite a relevant reference for serotonin in the action of Lithium. However, another citation is needed because there is direct biochemical evidence for the action of Lithium on serotonin. The authors are required to add the relevant reference, Neuropharmacology. 1982 Jul;21(7):671-9. doi: 10.1016/0028-3908(82)90010-7. PMID: 6181426.
We thank the reviewer for directing our attention to the publication linking Lithium to the serotonin pathway. We have now added the this reference on the serotonin link to Lithium action in the MS.
Reviewer 2 Report
Comments and Suggestions for Authors
Killick. et al. perform a transcriptional characterization of rat-derived neuronal cells after acute exposure to Lithium, with a further potential drug repurposing analysis and exploratory analysis in a large cohort of psychiatric medication patients. The paper is interesting but requires a little work in writing and highlighting the results.
Introduction:
Line 73: Assuming neuronal interactions primarily drive the therapeutic effects. Even when there are still some gaps in the cellular background for bipolar disorder, please justify the use of neuronal cells instead of other cells (PMID: 39107272, PMID: 36307327, PMID: 38281050, PMID: 37217515)
Line 75 to 99:
Are these results from the current manuscript? If so, this should be included in that section or at the end of the introduction.
Instead of adding a description of the current study in the introduction, please add the aim of the current research and expand the literature review of the current transcriptional changes in MDD (PMID: 39265803, PMID: 28825715), BP (PMID: 31118907, PMID: 23360497, PMID: 38351171) and Lithium (PMID: 36653674, PMID: 31652432, PMID: 31797941, PMID: 33498969,, PMID: 27867936, PMID: 32251271).
Methods.
Line 117: How were 10 mM and cells cultured for a further 2 h determined?
Line 133: What was the model used for DEG analysis, covariates, if any, and batch effects?
Line 142: For the MDD and BD analysis, how did you handle the heterogeneity in brain tissue and the potential effects of different arrays? Also, a supplementary table with more descriptive information for each study should be added.
Line 158: Could you compare the prescription of digoxin with those outside a psychiatric medication? Which psychaitric medicaitons were considered and why, add a supplemenatry table describing this.
Line 167: Please low the lenguage of your descriptions, for example "elucidate" is a strong word given the analyzed methodology.
Line 175: I think this is important and should be added in the introduction, maybe with the aim of the manuscript and not until here.
Figure 1. Please represent this better. Maybe a network or something similar? Also, please add colors if this gene has been previously reported to be responsive in Li exposure.
Figure 2. It is only showing MDD, what happen with the results for all the GEO data analyzed? Please build a figure with this information and add the genome-wide results as supplementary.
Line 254: Table XXX, please adjust. Add more information for all the Tables footnotes, and do not use a screenshot of the tables.
Line 257: Add the logistic regression information in methods and a replication sample.
Conclusion.
Add limitations of the study.
How will you propose to use CAT in the context of MDD and BD, given the high toxicity of these drugs (example: PMID: 39265879)
Also, discuss the context of the cell used in this experiment (Rat compared to humans)
There are several typos in the manuscript; please review it extensively.
Author Response
Please see our comments in italics below.
Killick. et al. perform a transcriptional characterization of rat-derived neuronal cells after acute exposure to Lithium, with a further potential drug repurposing analysis and exploratory analysis in a large cohort of psychiatric medication patients. The paper is interesting but requires a little work in writing and highlighting the results.
Introduction:
Line 73: Assuming neuronal interactions primarily drive the therapeutic effects. Even when there are still some gaps in the cellular background for bipolar disorder, please justify the use of neuronal cells instead of other cells (PMID: 39107272, PMID: 36307327, PMID: 38281050, PMID: 37217515)
We thank the reviewer for asking for this important clarification. Both BP and CDD are disorders of the cortical regions of the brain. Thus, we chose to investigate drug activity in the context of neuronally enriched mixed primary cortical cultures as this provides the best in vitro model system of the cerebral cortex. Specifically, these cultures better mimic the intact brain than any cell lines and as grown in vitro allow the Li treatment to be applied at a precise concentration and time frame. We have added a statement to this effect in the Introduction.
“We reasoned that these primary neuronal cultures provide a better approximation to the disease context drug activity than cell lines and, in contrast to in vivo assays, allow for a precise control of dosing and treatment time.”
Line 75 to 99:
Are these results from the current manuscript? If so, this should be included in that section or at the end of the introduction.
Instead of adding a description of the current study in the introduction, please add the aim of the current research and expand the literature review of the current transcriptional changes in MDD (PMID: 39265803, PMID: 28825715), BP (PMID: 31118907, PMID: 23360497, PMID: 38351171) and Lithium (PMID: 36653674, PMID: 31652432, PMID: 31797941, PMID: 33498969,, PMID: 27867936, PMID: 32251271).
We are sorry that the wording is confusing. We have clarified that these results are indeed a part of the work presented in the MS. We have clarified the text along these lines. We also thank the reviewer for pointing out the additional references in relation to MDD, BP and Li. We have now included references to these in the MS.
“Thus, our investigation will furnish a valuable addition to the existing studies of the Li driven transcriptional landscape[26-31].”
“To investigate the therapeutic aspects of the gene expression changes driven by Li we analysed transcriptional data on two conditions for which Li is commonly prescribed, major depressive disorder (MDD) and bipolar disorder (BP). There is a wealth of gene expression data on both these conditions: MDD[32-36] and BP[35, 37-44]. We report here that our acute Li profile (ALP) shows a significant negative correlation with transcriptional profiles corresponding to both MDD and BP derived from post-mortem brain samples.”
Methods.
Line 117: How were 10 mM and cells cultured for a further 2 h determined?
Li was used at 10 mM as this is a widely used sub-toxic acute dose which has rapid effects on targets including GSK3 and which mimics Wnt signalling activity (PMID: 8994831) and impacts the NR4A family. (PMID: 18727708)
“This dose is a widely used sub-toxic acute dose which has rapid effects on targets including GSK3 and which mimics Wnt signaling activity[7] and impacts the NR4A family[47].”
Line 133: What was the model used for DEG analysis, covariates, if any, and batch effects?
We have clarified our DEG methodology
“In the metanalysis of the disease profiles we covaried with brain region, sex and age where these features were provided.”
Line 142: For the MDD and BD analysis, how did you handle the heterogeneity in brain tissue and the potential effects of different arrays? Also, a supplementary table with more descriptive information for each study should be added.
Please see comment above. We have clarified that our DEG analysis was performed with appropriate covariates to control for confounding factors. We have provided links to the NCBI GEO repositories for the individual studies, and these give clear descriptions of the sample characteristics.
Line 158: Could you compare the prescription of digoxin with those outside a psychiatric medication? Which psychaitric medicaitons were considered and why, add a supplemenatry table describing this.
We have supplied a table with the psychiatric medication lists. We have also provided ATC codes for simple reference https://atcddd.fhi.no/. We have added the URL to the supplementary table. This list comprises to total prescription set for psychiatric medication in Norway.
Line 167: Please low the lenguage of your descriptions, for example "elucidate" is a strong word given the analyzed methodology.
We have rephrased this: “shed light on”
Line 175: I think this is important and should be added in the introduction, maybe with the aim of the manuscript and not until here.
We have added some further motivation for acute treatment in the Introduction, see comment above.
Figure 1. Please represent this better. Maybe a network or something similar? Also, please add colors if this gene has been previously reported to be responsive in Li exposure.
To enhance the representation of the data we have now included a volcano plot with a colour scale based on the significance Z score. We feel that this gives an intuitive picture of the overall transcriptional profile. Our profile is based on acute Li treatment and as such shares little with published Li driven profiles. We therefore cannot make a direct comparison here and have instead described some of the most significantly perturbed genes in the text.
Figure 2. It is only showing MDD, what happen with the results for all the GEO data analyzed? Please build a figure with this information and add the genome-wide results as supplementary.
The comparison is with the summary MDD and BP profiles that are combinations of all the profiles described in the Methods section. We have clarified this in the Figure legend.
“The topmost Li regulated genes are shown together with the expression changes in the combined meta-analysis profiles corresponding to MDD (left) and BPD (right) showing a clear reversal. Of the 40 most regulated genes due to Li treatment 32 in 40 and 22 in 32 are regulated in an opposite sense in MDD and BPD.”
Line 254: Table XXX, please adjust. Add more information for all the Tables footnotes, and do not use a screenshot of the tables.
We thank the reviewer for noticing this error which we have now rectified. Please note that we will provide Table formatted according to publication guidelines upon publication.
Line 257: Add the logistic regression information in methods and a replication sample.
We have now added a note in Methods.
“Prescription associations were assessed through logistic regression with appropriate covariates using the R glm package.”
Conclusion.
Add limitations of the study.
How will you propose to use CAT in the context of MDD and BD, given the high toxicity of these drugs (example: PMID: 39265879)
We have now added a sentence to this effect: “As with any proposed repurposing, the relative safety profiles of the existing and candidate drugs have to be carefully considered before deployment.”
Also, discuss the context of the cell used in this experiment (Rat compared to humans)
Dissociated cortical cultures are a good, widely used model system but differ substantially from the intact brain. We have expanded on our choice of neuronal cell cultures for studying acute Li treatment driven transcription in the Introduction.
Reviewer 3 Report
Comments and Suggestions for Authors
In their manuscript "Transcription driven repurposing of CTS for Li treatment of severe depression" the authors screened publicly available databases to identify CTS that elicit the same transcriptional signature as Li in the early stages of treatment. They have identified several candidates that might be able to provide the same therapeutic benefits as Li without the side-effects.
The study is essentially correlative and therefore speculative. Considering that the authors emphasise this point and are very clear about the limitations of the current study, I think it provides interesting new candidates to test in directly in animal models of the condition of interest.
My only comment is regarding the choice of early onset genes. I understand the rational but I would like the authors to expend on how relevant the earliest changes in genes expression following Li treatment are to the clinical benefits if they do not appear for another 6 months? Have the authors also considered the late transcriptional changes? Is there any correlation with the CTS profiles?
Minor comments: there are a few typos across the manuscripts (e.g. "Table XXX" line 254). Nothing a thorough proof reading won't pick up.
Author Response
Please see our comments in italics below
In their manuscript "Transcription driven repurposing of CTS for Li treatment of severe depression" the authors screened publicly available databases to identify CTS that elicit the same transcriptional signature as Li in the early stages of treatment. They have identified several candidates that might be able to provide the same therapeutic benefits as Li without the side-effects.
The study is essentially correlative and therefore speculative. Considering that the authors emphasise this point and are very clear about the limitations of the current study, I think it provides interesting new candidates to test in directly in animal models of the condition of interest.
My only comment is regarding the choice of early onset genes. I understand the rational but I would like the authors to expend on how relevant the earliest changes in genes expression following Li treatment are to the clinical benefits if they do not appear for another 6 months? Have the authors also considered the late transcriptional changes? Is there any correlation with the CTS profiles?
Our choice to identify the earliest wave of transcriptional effects of Li was predicated on the direct effects of Li being obscured by subsequent waves of transcription especially if its targets are themselves DNA binding gene transcription factors, which our data show is indeed the case, among the top hits being the whole NR4A family of Nuclear Receptor transcription factors all strongly associated with dopaminergic neurons. Our initial aim was to identify those initial targets and doing that then revealed the association which many other studies, using much longer, chronic time course treatments have missed.
Minor comments: there are a few typos across the manuscripts (e.g. "Table XXX" line 254). Nothing a thorough proof reading won't pick up.
We thank the reviewer for pointing out this error which we have now corrected.
Round 2
Reviewer 2 Report
Comments and Suggestions for Authors
Thank you for addressing my comments.